# Breast Sarcomas—How Different Are They from Breast Carcinomas? Clinical, Pathological, Imaging and Treatment Insights

**DOI:** 10.3390/diagnostics13081370

**Published:** 2023-04-07

**Authors:** Iulian Radu, Viorel Scripcariu, Andrian Panuța, Alexandra Rusu, Vlad-Adrian Afrăsânie, Elena Cojocaru, Maria Gabriela Aniței, Teodora Alexa-Stratulat, Cristina Terinte, Cristinel Florin Șerban, Bogdan Gafton

**Affiliations:** 1First Surgical Oncology Unit, Department of Surgery, Regional Institute of Oncology, 700483 Iasi, Romania; 2Department of Surgery, Faculty of Medicine, “Grigore T. Popa” University of Medicine and Pharmacy, 700115 Iasi, Romania; 3Clinic of Plastic and Reconstructive Microsurgery, Emergency Clinical Hospital “Sf. Spiridon”, 700111 Iasi, Romania; 4Department of Medical Oncology, Regional Institute of Oncology, 700483 Iasi, Romania; 5Department of Oncology, Faculty of Medicine, “Grigore T. Popa” University of Medicine and Pharmacy, 700115 Iasi, Romania; 6Department of Morphofunctional Sciences I—Pathology, “Grigore T. Popa” University of Medicine and Pharmacy, 700115 Iasi, Romania; 7Department of Pathology, Regional Institute of Oncology, 700483 Iasi, Romania; 8Department of Pathology, “Dr. C. I. Parhon” Clinical Hospital, 700503 Iasi, Romania

**Keywords:** breast sarcoma, breast carcinoma, diagnosis, treatment

## Abstract

Breast sarcoma (BS) is a very rare and poorly studied condition. This has led to a lack of studies with a high level of evidence and to low efficacy of current clinical management protocols. Here we present our experience in treating this disease in the form of a retrospective case series study including discussion of clinical, imaging, and pathological features and treatment. We also compare the main clinical and biological features of six cases of BS (phyllodes tumors were excluded) with a cohort of 184 patients with unilateral breast carcinoma (BC) from a previous study performed at our institution. Patients with BS were diagnosed at a younger age, presented no evidence of lymph node invasion or distant metastases, had no multiple or bilateral lesions, and underwent a shorter length of hospital stay versus the breast carcinoma group. Where recommended, adjuvant chemotherapy consisted of an anthracycline-containing regimen, and adjuvant external radiotherapy was delivered in doses of 50 Gy. The comparison data obtained from our BS cases and the ones with BC revealed differences in diagnosis and treatment. A correct pathological diagnosis of breast sarcoma is essential for the right therapeutic approach. We still have more to learn about this entity, but our case series could add value to existing knowledge in a meta-analysis study.

## 1. Introduction

Breast sarcoma (BS) defines a heterogeneous group of non-epithelial malignant tumors originating from mesenchymal mammary gland tissues. These tumors present unique histologic and cytogenetic peculiarities, specific clinical implications, and evolutionary patterns that are categorically distinct from the characteristics of breast carcinomas [1]. BS exhibits accelerated growth, more aggressive behavior, poorer response to adjuvant treatment, and higher tendency to both local recurrence and systemic metastasis [2,3,4]. In this paper we reviewed the literature and described a case series from our institution on this rare entity, focusing on the features and management of breast sarcomas and the differences compared to breast carcinomas.

## 2. Materials and Methods

Of the 4046 patients who underwent surgery for malignant tumors of the breast between May 2012 and October 2021 in the surgical department of the Regional Institute of Oncology, Iasi, Romania, only six (0.15%) female patients were diagnosed with primary sarcomas of the breast (PBS). The small number of cases available does not allow the use of analytical statistic methods (a minimum of 30 cases are needed to obtain reliable data). Therefore, our research was conducted in a retrospective, descriptive manner, and is designed as a case series study. Given their dual morphological nature, namely a combination of epithelial and stromal components, we decided to exclude phyllodes tumors from our study because they do not fit the histological definition of a sarcoma.

Written consent regarding the usage of clinical and pathological data of the patients included in our study was obtained and approved by the ethics committees of our hospital (Regional Institute of Oncology, Iasi) and the Grigore T. Popa University of Medicine and Pharmacy, Iasi, Romania.

The inclusion criteria for the current study were a pathologically confirmed diagnosis of breast sarcoma and no personal history of simultaneous other malignancies. The following characteristics were evaluated: tumor size, histological type, differentiation grade, nodal status, stage, imaging and pathological features, type of surgery, and multidisciplinary tumor board decisions. In all cases, immunohistochemical methods were used, since without them a reliable diagnosis of sarcoma is practically impossible to establish. All of these features were assessed according to the classifications of the 8th Edition of the American Joint Committee on Cancer/Union for International Cancer Control (AJCC/UICC) Staging for Soft Tissue Sarcomas of the Extremities and Trunk [5].

In order to compare the results obtained in the studied BS group, we used data from a previous study conducted in our institution covering a cohort of 184 patients with unilateral breast carcinoma (BC) who underwent surgical treatment during the same period as our main group of cases. 

## 3. Results

### 3.1. Age of Patients

By analyzing the data strings, namely the age at diagnosis of patients with PBS, we obtained an average age of 57 and a median of 59 (Table 1).

### 3.2. Residence Area

In our study, PBS patients were predominantly urban inhabitants—66.6% versus 33.3% rural (Table 1).

### 3.3. Laterality of Breast Tumors

In patients with PBS, 50% of breast tumors were localized in the right breast and 50% in the left breast (Table 1). No cases with bilateral or one-sided multiple tumors were identified in the PBS group. 

### 3.4. Length of Hospital Stay (LOS)

The average length of hospitalization was 6.3 days, and the median value was 6.5 (Table 1).

### 3.5. Lymph Node Invasion Rates (N Status) and Distant Metastases (M Status)

Our PBS cases had no lymph node invasion or distant metastases, although in several cases clinical examination or/and imaging methods suggested the presence of adenopathy or distant secondary lesions (confirmed by high-performance imaging and/or pathology exam) (Table 2 and Table 3).

### 3.6. Imaging

Our patients were examined using one or both conventional methods of imaging common for breast pathology—mammography and breast ultrasonography. High-performance imaging, especially computer-tomography scans, was used to exclude distant metastases. A detailed description of the imaging features of each case is presented in Table 2. 

### 3.7. Surgical Treatment

Half of the PBS cases (3 cases/50%) underwent conservative surgical procedures such as lumpectomy and quadrantectomy, while in three cases (50%), due to the clinical and imagistic appearance of the locally advanced primary tumor and adenopathy or the aggressiveness of the initial histological diagnosis, the patient underwent extended surgery (total mastectomy) without lymphadenectomy or involving partial (level 1 axillary node cleansing) or complete lymphadenectomy (Madden-type modified radical mastectomy) (Table 3).

### 3.8. Histological Types

Pre- and intraoperative establishment of the pathologic diagnosis was a challenge for our multidisciplinary team. We performed core biopsies in four patients with BS and three extemporaneous intraoperative examinations of the specimen. It is necessary to mention that one case underwent both pre- and intraoperative morphologic examination; in other words, the same patient had a core biopsy and an extemporaneous examination. Misdiagnosis in our research was the rule rather than the exception because, surprisingly, none of the patients received a correct diagnosis using core biopsy or/and extemporaneous exams The only value of these methods was to suggest a malignant tumor, which led to surgical treatment. Specifically, two cases of BS were interpreted as metaplastic carcinoma, two as a phyllodes tumor, one as medullary-type breast carcinoma and one was interpreted vaguely as a malignant tumor of the breast (Table 2). 

Definitive pathological diagnoses were established based on postoperative resection specimens. We ascertained definitive pathological diagnoses using immunohistochemistry methods. We found two cases of leiomyosarcoma of the breast and one case each of pleomorphic undifferentiated sarcoma, periductal stromal sarcoma, myxofibrosarcoma, and primary breast osteosarcoma (Table 3).

Microscopical aspects of some of our cases (leiomyosarcoma of the breast, myxofibrosarcoma, and primary breast osteosarcoma) are captured and illustrated in Figure 1, Figure 2, and Figure 3, respectively.

We diagnosed two cases of leiomyosarcoma of the breast and one case each of pleomorphic undifferentiated sarcoma, periductal stromal sarcoma, myxofibrosarcoma, and primary breast osteosarcoma.

In our study, two cases were initially diagnosed as metaplastic carcinoma and one case as medullary carcinoma. Therefore, it is important to make a distinction between BS and BC. It is impossible to differentiate microscopically between these two entities, as BS mimics many malignant epithelial and mesenchymal tumors [6,7]. A differential diagnosis should also include sarcomatoid carcinoma, carcinosarcoma, fibromatosis and fibrous histiocytoma [8]. Immunohistochemistry is mandatory to distinguish PBS from non-mesenchymal malignant tumors.

Regarding mammary leiomyosarcoma (Figure 1), the description was typical for a slow-growing malignant tumor with smooth muscle differentiation. Microscopically, the lesion had infiltrative edges and fascicular growth patterns. The malignant cells had eosinophilic cytoplasm and elongated nuclei, pronounced atypia, and high mitotic activity. In terms of immunohistochemistry, this neoplasm was positive for SMA, desmin, and caldesmon, which accounted for the differential diagnosis of metaplastic carcinoma as well as for other fusiform cell lesions [9].

Macroscopically, low-grade myxofibrosarcoma (Figure 2) had a gray color in the section and presented mucinous and fibrous areas. Microscopically, infiltrative edges and architecture were found, consisting of two distinct areas, one richly collagenized and hypocellular and another hypercellular with myxoid stroma. This type of tumor can also appear as short fascicles, whirlpools, or curved vessels. Malignant cells were monomorphic and mitotic activity was low, in contrast to the aggressiveness of this tumor. In terms of immunohistochemistry, low-grade myxofibrosarcoma was positive for MUC4, BCL-2, CD99, and vimentin. Differential diagnosis was made with epithelioid sclerosing fibrosarcoma, solitary fibrous tumor, low-grade myxofibrosarcoma, breast-type myofibroblastoma, borderline or malignant phyllodes tumor, and myxoid liposarcoma [10]. 

When establishing the diagnosis of extraosseous osteosarcoma (Figure 3), it is necessary to exclude bone origin and the presence of epithelial structures [11]. Macroscopically, the lesions were large and relatively well-defined, of hard consistency, and presented hemorrhage and necrosis with focal calcifications. Microscopically, osteosarcoma had infiltrative edges, while the shape of malignant cells was fusiform, similar to osteoblasts or osteoclasts. The presence of malignant bone features was essential for our diagnosis. In terms of immunohistochemistry, the SATB2 marker was positive and proved to be useful when the osteoid nature was difficult to identify. The main differential diagnosis considered was metaplastic carcinoma [11,12].

### 3.9. Adjuvant Treatment

All cases (6/100%) received adjuvant external radiotherapy in doses of 50Gy, and half underwent adjuvant chemotherapy (Table 3).

## 4. Discussion

### 4.1. Epidemiology 

According to Moore et al., the incidence of BS is 17 cases per million in women. Today’s worldwide prevalence of BS among all mammary malignancies is reported as less than 1%. As reported in one of the largest epidemiological studies on this subject (USA, Mayo Clinic, period from 1940 to 1999) by Adem et al., only 18 cases of BS were diagnosed out of 27,881 patients with mammary neoplasia, meaning an extremely low value (0.0006%) of the discussed index [4]. In our study, the incidence was 0.0014, similar to data from the literature. Breast localization accounts for approximately 5% of all sarcomas. Like breast carcinoma (BC), PBS is diagnosed almost exclusively (97.6% of cases) in women. Similarly, all patients included in this study (100%) were females. 

In comparison to BC, BS shows a distinctive histological/immunohistochemical (IHC) profile and a particular clinical-evolutive pattern characterized by accelerated tumor growth, aggressive biological behavior, greater tendency of both local recurrence and systemic metastases, and poor response to adjuvant treatment methods [2,3,4,13,14,15,16,17,18,19,20]. 

The extreme rarity of PBS leads to a lack of studies with high-level evidence, such as meta-analyses and systematic reviews on the disease, and as a result, to low efficacy of current clinical management protocols. Most of the studies on PBS (including our own) are designed in a retrospective manner, being classified as case series studies with a low impact on the knowledge base in this field. In Table 4 we summarize the main studies that analyzed patients with PBS, and the number of BS cases described [2,3,4,16,21,22,23,24,25,26,27,28,29,30,31,32].

### 4.2. Etiology

BS can be categorized as primary or secondary when considering the etiology of the disease, as follows:
Primary breast sarcoma (PBS) occurs de novo in the mammary parenchyma, the specific risk factors for this type of disease remaining unknown. Several genetic syndromes such as Li-Fraumeni syndrome, familial polyposis, or type-1 neurofibromatosis seem to increase the risk of developing sarcomas in general without being specific for mammary localization.Secondary breast sarcoma (SBS) occurs as an iatrogenic side effect of chest irradiation or in the background of chronic lymphedema.

To be considered as a secondary form, BS must have different histology from the initial lesion (usually BC) and must occur after a latency period in the irradiated territory, with the peak incidence at 5–10 years after radiotherapy. There are cases of SBS developing in the background of external irradiation for other malignant diseases regionally distant from the mammary gland, such as cervical cancer and non-Hodgkin’s lymphoma. Regarding the etiology of SBS, the role of exposure to environmental factors such as arsenic, chloride, or vinyl is cited by some authors. Breast angiosarcoma is the most common histologic type of SBS. It appears in the same site as previously treated BC in 73% of cases, and is a specific marker for radio-induced BS. Follow-up for these patients should not ignore discoloration and/or thickening of the skin, which may be the first signs of an early SBS. It is noticeable that similar incidence and clinical behavior to BS is shared by phyllodes tumors of the breast; nevertheless, its inclusion in the classification of BS is controversial considering the tumor’s mixed composition (epithelial and stromal tissues), which does not match the pathological definition of sarcoma. This point of view is also supported by the present study [19,33,34,35].

### 4.3. Clinical Data

We compared the PBS group with data published in another article [36] about a cohort of 184 patients with unilateral BC from our institution database. 

The average age for PBS cases was 57 versus 61.3 years old for BC ones, and the median age was 59 for BS patients versus 62 for BC cases (Table 1). The comparison of these series of values suggests the younger age of the participants in the PBS versus the BC group. Considering the low number of cases in the PBS series, we cannot produce a statistically valuable interpretation of these results. Nevertheless, larger studies on PBS show that the average age of the cases is approximately 49.5 years old versus 62 for BC cases [19]. 

In our study, both PBS patients and those from our control group (BC) have shown a prevalence of urban inhabitance—66.6% versus 33.3% for PBS cases and 71.2% versus 28.8% for BC cases (Table 1). In the reviewed literature, similar ratios of rural/urban living are found for patients with both epithelial and nonepithelial malignancies of the breast [9,21]. 

In patients with PBS, 50% of breast tumors were found in the right breast and 50% in the left breast (Table 1). In the BC group the result was nearly similar, with these patients showing a slightly higher “preference” (54.3%) for the left breast versus the right one (45.7%). Usually, PBS appears as a one-sided (a bilateral form of PBS is an exceptional finding), painless, and firm mass, having a larger size at the time of diagnosis (5–6 cm), and accelerated growth compared to BC cases. In most cases, the tumor develops as a single lesion and progressively invades glandular structures. On the other hand, bilateral and multicentric multifocal tumors are quite often found in BC patients [37], while multiple foci or bilateral lesions are extremely rare. For example, S. Al-Salam et al. first described bilateral primary angiosarcoma of the breast in 2012 [38].

The simple comparison of the average length (BS—6.3 versus 9.6 days for BC) and median value (BS—6.5 versus 9 days for BC) of hospital stay in the study groups reveals a slight prevalence of LOS in BC (Table 1). This result cannot be interpreted as a statistically significant result due to the small cohort, but can be explained by the prevalence of less extensive surgical procedures (without regional lymphadenectomy) performed in BS cases. Our BC patients mainly underwent radical surgery, which is more complex and had a longer postoperative recovery period and greater risk of early complications [39]. As long as two decades ago, it was reported that LOS in patients with BC decreased from 10–14 days to 5–7 days [40,41,42,43,44]. One of the major factors for the decrease in LOS was the trend towards breast-conserving surgery (BCS) instead of mastectomy.

### 4.4. Staging

Regarding BS, the American Joint Committee on Cancer (AJCC) TNM staging system considers the histological grade of tumors as a stage-determining criteria in addition to tumor size, lymph node metastases, and distant metastases [5].

Due to the haematogenic route of metastasis specific to connective tissue malignancies, invasion of the lymph nodes is rare, but when it is present it changes the patient’s prognosis dramatically. Similar to distant metastasis positivity (M1 status), N1 status in BS categorizes a case with any primary tumor dimension (T status) as a stage IV case [10]. Our BS cases had no lymph node invasion or distant metastases (Table 2 and Table 3). In contrast, the BC control group showed a very high level of N+ status (43.1%) and a significant percentage of M1 cases (9.78%). In the reviewed literature, lymphatic spread was uncommon and axillary lymph node involvement was not a frequent finding [45,46]. As for soft tissue sarcomas of other sites, metastases from primary breast sarcoma typically occur hematogenously, involving the lungs, bone marrow, and liver [8].

### 4.5. Imaging

Because breast sarcoma is rare, analysis of its imaging characteristics has been limited [8]. In this retrospective case series, imaging features were analyzed, but no specific trends could be observed; rather, they showed features of breast carcinoma.

In contrast to our study, findings in some retrospective analyses indicate that primary breast sarcomas present mammographic and sonographic imaging features that are different from those of typical infiltrating ductal carcinoma [47,48]. The same studies found the majority of breast sarcomas to be noncalcified oval masses with indistinct or circumscribed margins in mammography. However, mammography is not specified for diagnosis, as calcification in PBS is rare [47]. 

Therefore, the usual breast imaging methods are not very relevant for the diagnosis of PBS. In some cases, MRI may be useful in suggesting the sarcomatous nature of the tumor. In order to personalize a treatment plan for a patient with BS, it is mandatory to evaluate the eventuality of distant metastases using conventional and high-performance imaging methods to scan the most frequently affected organs [14,49].

### 4.6. Diagnosis

The most accurate method for establishing a diagnosis of BS remains the core biopsy with immunohistochemical (IHC) analysis of specific cytokeratins. Fine needle aspiration (FNA) has practically no value in diagnosing BS because of false negative or irrelevant results. If the results of a core biopsy fail to deliver a final diagnosis (i.e., spotty or doubtful staining), then an incisional biopsy is a reasonable choice before deciding on a surgical procedure. In our study, immunohistochemistry was a major input in obtaining the right diagnosis. 

As in the case of any rare tumor, PBS should be referred to a sarcoma reference centre to increase overall survival by investigating clinicopathological features and taking a multidisciplinary approach. This recommendation is reinforced by the high rate of misdiagnosis reported in the literature for soft tissue sarcomas, which was also observed in our study in primary breast sarcomas. Between 25% and 40% of patients with soft tissue sarcomas are misdiagnosed, and these proportions have remained surprisingly unchanged over time despite the development of new techniques that facilitate pathological diagnosis [50,51,52,53]. However, a relatively recent retrospective study led by Ray-Coquard et al. found lower discordance rates than previously reported: 8% for major discordances (benign versus sarcoma, or sarcoma versus non-mesenchymal tumor) and 35% for minor discordances (sarcoma with different histopathological subtypes or grades) [54]. Accurate diagnosis remains difficult to obtain by non-specialist pathologists and in institutions that do not have access to resources like immunohistochemistry and molecular biology. Unfortunately, these misdiagnoses have unfavourable therapeutic implications for patients.

### 4.7. Histological Types

The most common histologic types of PBS, in order of frequency, are presented in Table 5 [19,37,49,55,56,57,58,59].

### 4.8. Treatment

In our study group of PBS, breast-conserving surgery and radical surgery was performed in three cases (50%), two patients (33.3%) underwent radical modified mastectomy (with axillary lymphadenectomy), and one patient (16.6%) received a simple total mastectomy (without axillary lymphadenectomy). These proportions were at odds with data presented in similar studies. A review of all reported cases in the literature revealed that 73% of BS patients developed tumor recurrences after breast-conserving therapy [60]. Therefore, total mastectomy is considered the main treatment method for BS, as for other rare histological breast cancer subtypes [61]. Most patients undergo radical (modified) mastectomy, but in carefully selected cases with relatively small tumors, breast-conserving procedures are performed. Systematic regional lymphatic dissection is not indicated if there is no clinical data to suggest lymph node invasion because of the low rate of lymph node invasion in PBS. On the other hand, all our BC patients underwent Madden-type modified radical mastectomy. However, an adequate resection margin is the most important determinant of long-term survival in breast cancers.

Considering the histological variant of the tumor and other clinical and morphological factors of the specific case, complementary treatment methods such as radio- and chemotherapy are used in the management of BS. Our MDT decided to recommend adjuvant radiotherapy with or without adjuvant chemotherapy. Adjuvant chemotherapy consisted of an anthracycline-containing regimen (Doxorubicin and Ifosfamide/Cisplatin), while adjuvant external radiotherapy was delivered in doses of 50 Gy. However, the decisions were not based on clinical guidelines or any protocol treatment, because of the rarity of breast sarcomas. There are no prospective randomized trials to guide therapy. Several principles of treatment have been derived from small retrospective case reviews of breast sarcomas, as well as studies of soft tissue sarcomas of the extremities and chest wall, since there are similarities in clinical behaviour, histology, and prognosis.

In our study the main features that led to the recommendation of radiotherapy were conservative surgery, tumors larger than 5 cm, and high-grade tumors (G3). This principle was extrapolated from soft tissue sarcomas, but scarce data support this decision algorithm. The benefit of radiotherapy for disease-free survival is unknown, even though many retrospective studies have attempted to analyse its efficacy. McGowan et al. demonstrated that patients who received radiation doses above 48 Gy had a cause-specific survival (CSS) of 91% versus 50% in the group that did not receive radiotherapy, or received radiation doses below 48 Gy. Therefore, they recommend postoperative irradiation in doses of 50 Gy to the whole breast and doses of at least 60 Gy to the tumor bed. Johnstone et al. obtained similar results to those reported in the study mentioned above: 5-year disease-free survival was 68% versus 47% in the study conducted by McGowan et al., and overall survival was 66% versus 57%. Therefore, both concluded that adjuvant radiotherapy decreases the rate of locoregional recurrence and increases disease-free survival [3,31]. Some studies in the literature have noted that those who benefit the most from adjuvant radiotherapy are patients with high-grade tumors (2 or 3) and those with large tumors, with the greatest dimension exceeding 5 cm [2,23,62,63]. Specifically, one of the largest studies about PBS to date showed that adjuvant radiotherapy reduces the risk of death by 36% in T2N0M0 PBS, without having a benefit on overall survival in T1N0M0 tumors [62]. However, numerous other studies have failed to demonstrate a statistically significant difference between patients who underwent surgery and received adjuvant radiotherapy and those who did not [21,23,26,64,65]. 

In the absence of any specific evidence for PBS, soft tissue sarcoma guidelines dictate the chemotherapy approach. A prospective study demonstrated that neoadjuvant chemotherapy (three full-dose courses of an anthracycline plus ifosfamide full-dose regimen) had a positive impact on overall survival and relapse-free survival of high-risk STS, which may be assumed for breast sarcoma as well [66]. There are a few reported cases of PBS in the literature that support these data, but they seem to be more of an exception than the rule [67,68]. As for adjuvant chemotherapy, its role remains unclear. Gutman et al. revealed increased disease-free survival and improved overall survival for patients who received adjuvant chemotherapy, while Zelek et al. recommended it only for high-risk PBS (high-grade tumors exceeding 5 cm) [22,62]. The combination of doxorubicin and ifosfamide is used when adjuvant chemotherapy is indicated. A taxane-containing regimen, however, may be appropriate for patients with angiosarcoma previously treated with anthracycline-based chemotherapy. Palliative chemotherapy underlies the treatment of metastatic PBS. As in the treatment of other soft tissue sarcomas, the recommended regimen is generally based on anthracyclines (Doxorubicin, Epirubicin, or Liposomal Doxorubicin) and ifosfamide. The following can also be used as therapeutic agents: Gemcitabine, taxanes (Docetaxel, Paclitaxel), Dacarbazine, and Vinorelbine. However, most experts recommend treatment choice on a case-by-case basis, depending on the histological subtype and clinical features of each patient. Furthermore, radiotherapy may be used to relieve certain symptoms, and surgical resection of metastases may be considered [63]. In our research, the arguments for choosing adjuvant chemotherapy were the presence of aggressive histologies (fibromyxosarcoma, leiomyosarcoma, and osteosarcoma), high-grade tumors, and/or high proliferation rate. 

Clinical outcomes for localized disease are subject to considerable variations. Five-year overall survival is considered to be between 49% and 67% [3,4,16,22,34,62,69,70]. These ranges are due to the heterogeneity of histological subtypes and treatment protocols analyzed in each study. The most relevant prognostic factors identified are grade [26,64], tumor dimensions [23,63,69], and margin resection status [21,63,69,71]. However, osteosarcoma and angiosarcoma appear to be the histological subtypes with the poorest prognosis [63,72]. In contrast, fibrosarcoma and liposarcoma appear to have the best outcomes [63].

### 4.9. Surveillance

Surveillance of surgically treated PBS is based on two principles: in low-grade BS, many recurrences that occur are local, and in high-grade BS, distant metastases, mostly to lungs, are more common in the first two years [62,73]. Although there are no prospective studies, ESMO guidelines for soft tissue sarcoma recommend stratifying patients according to their estimated risk of locoregional recurrence. Thus, for intermediate- and high-risk sarcomas, monitoring is performed by physical examination, chest CT and breast MRI every 3–4 months in the first 2–3 years, biannually for up to 5 years, then annually. For low-risk sarcomas, evaluation is recommended every 6 months for the first 5 years, then annually [73].

### 4.10. Limitations and Possible Biases of the Study

The retrospective nature of the study;The low number of patients included in the study;Possible biases caused by human factors in errors in the completion of the database, clinical, imaging, and pathological evaluations of BS cases;The relative lack of large-scale studies (meta-analyses) on the chosen subject found in the literature for comparing the data obtained and, consequently, difficulties in drawing relevant conclusions.

## 5. Conclusions

Breast sarcoma is characterized by very low incidence and very high heterogeneity from the clinical, imaging, and histopathological points of view, with management protocols being totally different from those of well-studied breast carcinoma. The comparison of clinical and biological data obtained from our PBS group of cases and the ones with BC revealed differences in certain aspects. Namely, patients with sarcoma seem to be diagnosed at a younger age, with no cases of lymph node invasion or distant metastases found in the main studied group versus multiple N+ and M1 cases in the breast cancer (control) group, no multiple/bilateral lesions found in patients with breast sarcoma, and a shorter length of hospital stay for patients with sarcoma. Surgery, radiotherapy, and chemotherapy were recommended using general principles of sarcoma treatment and other retrospective data from the literature, without having any guidelines or strong-evidence data to lead our decisions. It is mandatory to mention that the small sample of cases with breast sarcoma that were treated in our unit does not allow us to make conclusions and extrapolate our findings as evidence-based data. Nevertheless, the results of our study are suitable to be summarized with those of other case series published in the field in order to perform meta-analysis studies. 

## Figures and Tables

**Figure 1 diagnostics-13-01370-f001:**
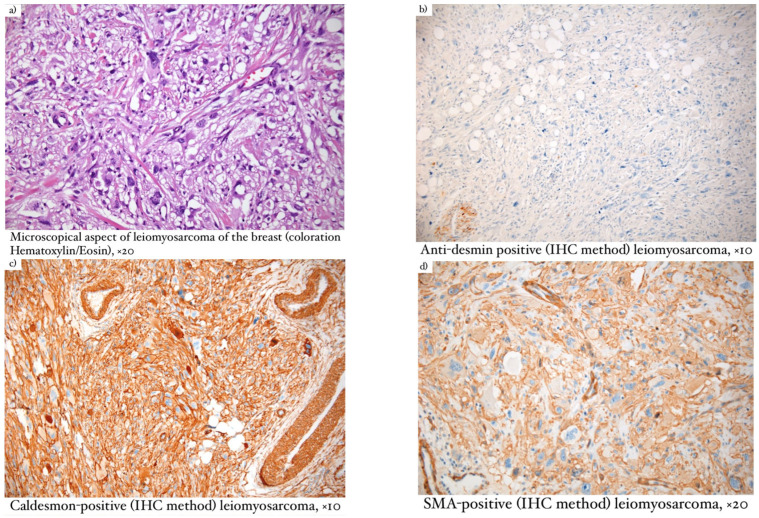
(**a**) Microscopical aspect of leiomyosarcoma of the breast (coloration Hematoxylin/Eosin), ×20; (**b**) Anti-desmin positive (IHC method) leiomyosarcoma, ×10; (**c**) Caldesmon–positive (IHC method) leiomyosarcoma, ×10; (**d**) SMA–positive (IHC method) leiomyosarcoma, ×20; (**e**) anti-muscle–positive (IHC method) leiomyosarcoma, ×20.

**Figure 2 diagnostics-13-01370-f002:**
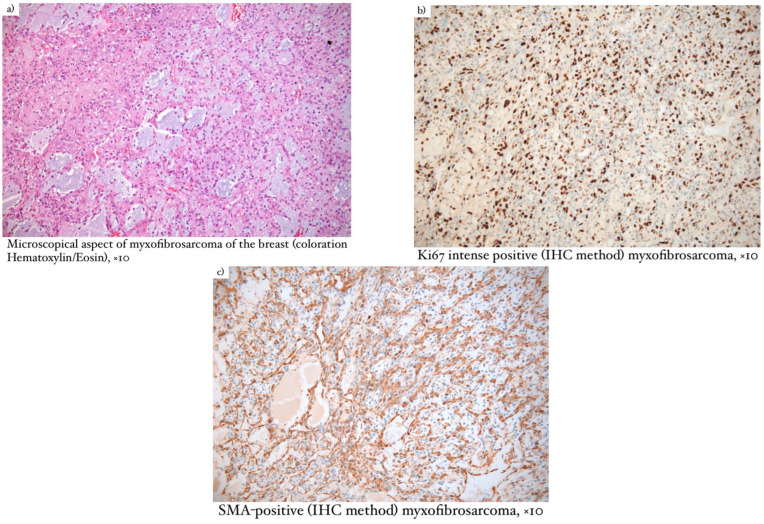
(**a**) Microscopical aspect of myxofibrosarcoma of the breast (coloration Hematoxylin/Eosin), ×10; (**b**) Ki67 intense positive (IHC method) myxofibrosarcoma, ×10; (**c**) SMA–positive (IHC method) myxofibrosarcoma, ×10.

**Figure 3 diagnostics-13-01370-f003:**
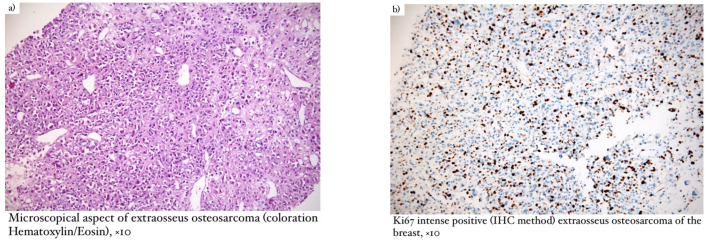
(**a**) Microscopical aspect of extraosseus osteosarcoma (coloration Hematoxylin/Eosin), ×10; (**b**) Ki67 intense positive (IHC method) extraosseus osteosarcoma of the breast, ×10; (**c**) SATB2 positive (IHC method) extraosseus osteosarcoma of the breast, ×10.

**Table 1 diagnostics-13-01370-t001:** Age, environment, tumor localization, and length of hospitalization in patients with breast sarcomas.

Patients	Age (Years)	Inhabitance Area	Laterality of PBS	Length of Hospital Stay (Days)
1	60	Urban	Right breast	7
2	49	Urban	Left breast	5
3	42	Urban	Right breast	6
4	69	Rural	Right breast	7
5	58	Rural	Left breast	8
6	64	Urban	Left breast	5
**Average value**	57	66.6% urban	50% left breast	6.3
**Median value**	59	33.3% rural	50% right breast	6.5

**Table 2 diagnostics-13-01370-t002:** Conventional imaging features and pre- and/or intraoperative pathological characteristics in patients with PBS.

Patient	Mammography	Ultrasonography of the Breast	Initial Histological Diagnosis Based on Extemporaneous Examination or Core Biopsy
1	In the upper inner quadrant of the right breast, a spiculated mass, of nodular density with a diameter of 20 mm, with micro-polylobate contours.	Cystic formation with a micro-polylobate contour of dimensions 13/11 mm, relatively well delimited, with vascularized parietal proliferations; absence of suspicious axillary adenopathies).	*Extemporaneous examination*: tumor proliferation with predominantly fusiform cells with sarcomatous aspect, with rich lympho-plasmocytic inflammatory infiltrate in the periphery of the nodule, aspects corresponding to a metaplastic carcinoma with fusiform cells.
2	Partially highlighted on the mediolateral oblique incidence is a delimited oval opacity of 30/20 mm; macrocalcifications with benign appearance.	A nodular lesion with partially clear contour, partially well delimited, heterogeneous, with hyperechogenic areas alternating with hypoechogenic areas inside and several internal vascular trajectories, elastographic score 2/3, dimensions ∼25/17 mm, located at ∼5 cm from the nipple, without suspected lymphonodules bilateral axillary.	*Extemporaneous examination* revealed the malignant nature of the formation but could not specify the histopathological type.
3	NA	In the lower-outer quadrant (LOQ) of the right breast, at a distance of 4–5 cm from the nipple and corresponding to the palpable nodule, a nodular lesion with maximum diameters of 31/20/30 mm, partially net contour, partially imprecise, partially vascularized, inhomogeneous hypoechogenic structure, with a suspicious appearance of malignancy; right axillary: ganglia with a benign appearance.	*Core biopsy:* Histological and IHC aspects (CD34—positive in most cells; Bcl2—positive in several of the cells; CKc, CK5, p63—negative; Ki67—positive in >50% of tumor cells) suggesting a tumor of the fibroepithelial type, possibly phyllodes tumor. *Extemporaneous examination.* A nodular formation of 3.5/2.7/3 cm—phyllodes tumor.
4	A nodule in the upper-outer-quadrant (UOQ) of the right breast, 7/6/6 cm (previously 25/17 mm), lobulate contour and macrocalcifications inside, as well as the presence of an opaque nodule 17/11 mm in the upper-inner-quadrant (UIQ).	In the UOQ—at a distance of 65 mm from the nipple, immediately subcutaneously, expansive formation with mixed, solid, and liquid echo-structure, impure liquid areas inside and solid component, vascularized, net lobulated contour; overall dimensions 64/43/55 mm, which also associates minimal perilesional edema; in UIQ—well-delimited solid nodule 21/11 mm, homogeneous, without vascular signal inside—probably benign appearance. Axillary adenopathies were not found.	*Ultrasound-guided core biopsy* suggested a phyllodes tumor, in favor of a malignant phyllodes tumor (stromal growth, frequent mitosis).
5	NA	NA	*Core biopsy*: breast carcinoma of the medullary type.
6	At the level of the deep UOQ, opacity with characters suspicious of malignancy, of high intensity, with discretely irregular contour, with a few small extensions and partially erased by overlapping with the adjacent glandular tissue, with dimensions of 25/25 mm; density asymmetry in the UOQ in the vicinity of the opacity described above; several diffusely distributed microcalcifications.	In the UOQ, hypoechogenic formation, partially net contour, partially imprecise, in some places microlobulated, intensely vascularized, hard elastographic, with dimensions of 22/16/26 mm—uncertain ultrasound aspect; axillary, supra- and left subclavicular—the absence of suspected lymphonodules. ACR-BIRADS score b.4c.	*Core biopsy:* corresponds to a malignant tumor proliferation with epithelioid allure, with nonspecific IHC phenotype (CKAE1/AE3, S100, EMA, MART1, Desmin, CD30, CD45—negative in tumor cells. CK19, CK34BetaE12, CK-CAM 5.2, MUM1, ER, PR, HER2, CD34, p63—negative. Ki67—positive 20–25%), the aspects are compatible with a metaplastic carcinoma, without epithelial component in the biopsied samples.

**Table 3 diagnostics-13-01370-t003:** Type of surgery, definitive pathological exam results and cancer committee recommendations in breast sarcomas.

Patient	Type of Surgery Performed	Definitive Diagnosis (Paraffin Block)	TNM Stage	Multidisciplinary Oncological Commission Recommendations
1	Total mastectomy	Pleomorphic undifferentiated sarcoma, pT1aNxM0—G3; the tumor formation with dermal-hypodermic development showed a proliferation with predominantly fusiform cells, organized in beams of varied orientation, focal storiform, along with areas with cells with epithelioid allure, frequent atypical mitosis (over 10/10 HPF). No tumor aspects were found at the level of the excision margins.	II	Adjuvant external radiotherapy DT = 50 Gy/25 fr/2 Gy/fr
2	Lumpectomy	Breast leiomyosarcoma, pT1aNx—G2. The rest of the excised breast parenchyma showed aspects of fibrocystic mastopathy with periductal fibrosis and apocrine metaplasia.	II	Adjuvant external radiotherapy DT = 50 Gy/25 fr/2 Gy/fr
3	Quadrantectomy	A nodular formation consisting of a biphasic: mesenchymal proliferation of fusiform cells with moderate density arranged in sleeves around ducts with epithelial proliferation without atypia. Marked cytonuclear atypia, nuclear pleomorphism and mitotic activity (10 mitosis/10 HPF), stromal proliferation with myxoid areas were found. The lesion was excised with margins of oncological safety in all plans. IHC profile indicated the diagnosis of periductal stromal sarcoma, pT1NxM0—G3.	IA	Adjuvant external radiotherapy DT = 50 Gy/25 fr/2 Gy/fr
4	Total mastectomy with level I axillary lymphadenectomy (adenopathies of 5–15 mm at the level of the axillary station I)	A mesenchymal tumor with an abundant myxoid component, the IHC profile is compatible with a fibromyxosarcoma—G3, pT2N0M0, L1V1Pn0. Differential diagnosis includes a myxoid liposarcoma or an extrascheletic chondrosarcoma myxoid.	IIIA	Adjuvant chemotherapy with Doxorubicin (75 mg/m^2^) and Ifosfamide (5 g/m^2^), followed by adjuvant external radiotherapy DT = 50 Gy/25 fr/2 Gy/fr
5	Modified Madden-type radical mastectomy (clinical exam: ulcerated tumor, axillary adenopathies)	The morphological and IHC (h-Caldesmon—positive, Desmin—positive, CD34—weakly positive) aspects correspond to a poorly differentiated leiomyosarcoma, pT2N0M0—G3. No metastasis was found in the lymph nodes. Large areas of tumor necrosis (less than 50%) and frequent mitosis (over 20 mitosis/10 HPF) noticed in the pleomorphic areas.	IIIA	Adjuvant chemotherapy with Doxorubicin (75 mg/m^2^) and Ifosfamide (5 g/m^2^), followed by adjuvant external radiotherapy DT = 50 Gy/25 fr/2 Gy/fr
6	Lumpectomy	The overall morphological evaluation of the operative part proves the existence of a malignant tumor with osteoblastic differentiation without epithelial component on the examined sections, thus meeting the criteria for primary breast osteosarcoma, T1NxM0–Gx.	IA	Adjuvant chemotherapy with Doxorubicin (75 mg/m^2^) and Cisplatin (100 mg/m^2^) followed by adjuvant external radiotherapy DT = 50 Gy/25 fr/2 Gy/fr

**Table 4 diagnostics-13-01370-t004:** Number of cases of breast sarcoma in the reviewed literature [2,3,4,16,21,22,23,24,25,26,27,28,29,30,31,32].

Authors	Number of BS Cases
Bousquet et al., 2007 [21]	103
Gutman et al., 1994 [22]	60
Barrow et al., 1999 [23]	59
Silver et al., 1982 [24]	50
Donnell et al., 1981 [25]	40
McGowan et al., 2000 [3]	46
Terrier et al., 1989 [16]	33
Pollard et al., 1990 [26]	25
McGregor et al., 1994 [27]	20
Adem C et al., 2004 [4]	18
Moore et al., 1996 [2]	17
Merino et al., 1983 [28]	15
Jalil et al., 1996 [29]	11
Ventrillon et al., 1992 [30]	4
Johnstone et al., 1993 [31]	4
Falconieri et al., 1997 [32]	2

**Table 5 diagnostics-13-01370-t005:** Histological types of breast sarcomas in order of frequency, and IHC markers used to determine each type [19,37,49,55,56,57,58,59].

Histological Subtype	Frequency	Immunohistochemistry Markers
Angiosarcoma	33%	CD31, CKAE1/3, CAM5.2, EMA
Stromal sarcoma	9.8%	CD34, BCL2, CKc, CK5, p63, Ki67
Undifferentiated pleomorphic sarcoma	7.6%	Vimentin, CD34, CK7, CKAE1/AE3, ER, PR, HER2neu, p63, S100, MelanA, BCL2, ki67
Leiomyosarcoma	7.5%	SMA, desmin, caldesmon, CD34, S100, CKAE1/AE3, BCL2, Ki67
Fibrosarcoma	6%	CD34, ER, PR, AR, CD99, desmin, GFAP, p63, SMA, CKAE1/AE3, BCL2, CD31, S100
Liposarcoma	5.5%	CD34, ER, PR, AR, CD99, Desmin, GFAP, p63, SMA, CKAE1/AE3, BCL2, CD31, S100
Osteosarcoma	4.5%	SOX10, vimentin, CD56, CD34, CK8/18, CKAE1/AE3, CD68, actyn, CD23, CD138, HMB-45, S100, ERG, SATB2
Chondrosarcoma	0.5%	Vimentin, S100, NSE, CD99, SYN, osteopontin
Kaposi’s sarcoma	<0.5%	Vimentin, CD31, CD34, SMA, S100, EMA
Low-grade fibromyxoid sarcoma and other rare types	<0.5%	EMA, SMA, CD34, desmin, S100, MUC4

## Data Availability

The data presented in this study are available on request from the corresponding author. The data are not publicly available due to institutional policy.

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
