# Peer review of "Breast Sarcomas—How Different Are They from Breast Carcinomas? Clinical, Pathological, Imaging and Treatment Insights"

_diagnostics, 2023, doi:10.3390/diagnostics13081370_

Round 1
Reviewer 1 Report
I belive that the paper is adequate to be published.
I would sugest 2 recomendations:
1-lines 152/153 the text does not makes sense; the authors should remove the words "the other" in the middle of the sentence
2-the authors should reserve the term "radical surgery" to refer to breast surgery that includes axillary lymphadenectomy; for several times, the authors use the term "radical surgery" to refer to total mastectomy, and that is not correct. In oncology, radical means the removal of the lymph nodes anatomically related to the primary tumour.
Author Response
We would like to begin by thanking you for your valuable contribution brought to our paper and for acknowledging our hard work. In the next lines, we present our revisions:
1- We have removed the words “the other” (lines 152/153)
2- We have revised the use of the term “radical surgery” (lines 113, 364)
Reviewer 2 Report
One. The more interesting part of the paper is the description of 6 new cases including the illustrations in Figures 1 to 3. We would suggest presenting these in more detail with scale bars, label, and reference to the cases as numbered in Tables 2 & 3. Most part of the description on lines 295 – 345 does not belong to the Discussion section but would better be part of the Results (3.8).
Two. As mentioned by the authors (line 306) immunohistochemistry is crucial for the diagnosis and typing of breast carcinomas. Is it possible to add to Table 5 the IHC markers that were used to determine the histological types?
Three. Could the authors give some more details about the position of phyllodes tumors and explain why these tumors do not fit the histological definition of sarcoma (line 54).
Four. Some words do not indicate what they are meant to: “imagistic” (line 103) is a literary movement; “overwise” (line 158) . What is “the other half” of “all cases” (line 152).
Author Response
We would like to begin by thanking you for your valuable contribution brought to our paper and for acknowledging our hard work. In the next lines, we present our revisions:
- We have moved the description concerning the histological types of BS from Discussion to Results (lines 295-345 now lines 157-207). We strongly agree with you that the most interesting part of the paper is the description of the cases. We have added the labels. However, we have discussed with the pathologist and we have asked for the images and the scale bars, but unfortunately they are not available due to administrative issues.
- We have added a column in Table 5 in which we mentioned the IHC markers used to determine the histological types.
- We have explained the reason why the authors do not believe phyllodes tumours match the definition of sarcoma (lines 54-56)
- We have revised the use of the terms “imagistic” (line 107), “overwise” (line 215), and “the other” (line 209).